# The Spatial Spillover Effect and Its Impact on Tourism Development in a Megacity in China

Yajun Cao and Jianguo Liu *

Tourism College of Beijing Union University, Beijing 100101, China; caoyajun2022@163.com
* Correspondence: liujianguo@buu.edu.cn

**Abstract:** By utilizing the tourism development data of Beijing for the period from 2010 to 2019, this study examined the spatial pattern distribution of tourism development in Beijing using the coefficient of variation and Moran's I index. In addition, the geographic detector method was employed to explore the impact of tourism resource investment, tourism reception facilities, and urban development level on the spatial pattern of tourism development. The results indicate that the spatial differences in tourism development in various Beijing districts are gradually expanding, mainly focusing on the differences between urban function expansion regions. The number of tourists shows a spatial distribution pattern including a core area, urban function expansion area, ecological conservation area, and new urban development area. The spatial correlation of tourism development increases gradually, and some parts show the spatial correlation form of low–high aggregation. Tourism resource investment, tourism reception facilities, and urban development level all play a significant role in promoting the spatial pattern of tourism development, among which the most obvious role is the interactive effect of tourism reception facilities, star-rated hotels, and openness. Therefore, to improve the development of Beijing's tourism industry, the government needs to pay attention to the differences in the expansion of urban functions, the degree of contact between regions, the number of tourism reception facilities, and the level of regional openness. The significance of this research is in promoting spatial governance, coordinated development among regions, and the high-quality development of tourism in Beijing, and laying down a foundation for the introduction of spatial collaborative governance policies in other megacities in China.

**Keywords:** Beijing; tourism development; spatial pattern; geographic detector





## 1. Introduction

The recent rapid development of China's economy has made its population and industries concentrated in modern cities with high levels of economic activity and advanced infrastructure [1]. In this manner, a new urban scale in China, called "megacities," was created. The Seventh National Population Census of the People's Republic of China officially identified Beijing, Tianjin, Shanghai, Wuhan, Shenzhen, Chongqing, Guangzhou, and Chengdu as megacities. As the highest-ranking central cities and the most densely populated economic areas in China's urban system, the development of megacities plays an important role in promoting and demonstrating the synergistic development of the urban system and the transformation of China's urban development model. However, the concentration of the population, industries, and factors of production have led to "big city disease," which is characterized by traffic congestion, environmental pollution, and shortages of public services [2]. Effectively treating this malady and improving overall production, living standards, and the environment in megacities by appropriately adjusting the spatial structure of cities requires empirical evidence and thorough study.

Tourism plays an important role in cities, and the spatial pattern of its development can to a certain extent promote changes in the concentration of urban resources, industries, and economic factors, making it an important element for promoting the evolution

of inter-regional spatial structures [3]. In addition, tourism promotes a certain level of regional spatial coordination and cooperation. The balanced growth of developed and underdeveloped areas is an important factor in adjusting the balance and coordinating development between areas [4]. Thus, the spatial pattern of tourism development plays a pivotal role in spatial governance and the coordinated development of megacities. The tourism industry in Beijing, a representative megacity in China, was one of the first to be modernized and internationalized; therefore, an in-depth study of the spatial pattern of tourism development in Beijing and its influencing factors is not only important for optimizing spatial coordination, developing regional coordination, and developing high-quality tourism in Beijing, but will also serve as a reference and provide guidance for the coordinated spatial governance of other megacities in China.

## 2. Literature Review

Recently, the spatial pattern of tourism development has become a popular topic in academic research. This topic focuses mainly on the spatial correlation, heterogeneity, spillovers and coupling of tourism development, as well as the factors influencing the spatial pattern of tourism development. (1) The spatial correlation of tourism development. The first law of geography holds that everything is spatially related. Therefore, the development of tourism also has spatial relevance and correlation. Scholars studying the spatial correlation of tourism development have used social network analysis, modified gravity models, and data envelopment analysis. They found that the tourism industry has gradually formed a hierarchical and spatially optimized overall network correlation structure over the course of its development [5], in addition to the double-core and multipoint structure [6], core edge structure [7], multithreading and densification [8], central urban clustering and peripheral diffusion [9,10] and other local network structures. In addition, factors such as economies of scale, tourism costs [11], and transportation systems [12–14] affect the spatial correlation structure of tourism development. Advanced mobile data technologies have enabled new research methods that use big data to study the spatial correlation of tourism development [15]. (2) The spatial heterogeneity of tourism development. The second law of geography holds that the isolation of space causes the differences between things, that is, heterogeneity. Similarly, the development of tourism also has spatial heterogeneity. In the study if this topic, scholars have mainly used research methods such as the coefficient of variation, Theil index, and ESDA model, and there are many different places targeted in the research of domestic and foreign scholars. Foreign scholars mainly focus on the impact of macro factors, such as the international situation, on the heterogeneity of tourism development, while domestic scholars mainly focus on the impact of micro factors, such as global warming [16,17], crisis events such as coronavirus (COVID-19) [18,19], and regional factors such as high-speed rail networks [20,21], tourism infrastructure [22,23], and tourism resource endowment [24–26]. (3) The spatial spillover of tourism development. With the development of spatial econometrics, scholars have also gradually realized that the development of tourism can not only have an impact on the development of a region, but also have a certain impact on the development of adjacent regions, that is, the spatial spillover effect. The research in this area mainly focuses on the self-spillover of tourism development and the impact of other factors on it. Among them, the development of tourism can show its own spillover to the economic development, regional coordination and balance, and economic efficiency of the surrounding areas [27,28]. Meanwhile, technical efficiency [29], industrial clustering [30], tourist mobility [31], and urbanization [32,33] also influence the spatial spillover effects of tourism development. (4) The spatial coupling of tourism development. Most of the studies on the spatial coupling of tourism development are from Chinese researchers, mainly due to China's collaborative development strategy. Research has focused on the coupling and synergy between transportation [34], ecology [35], urbanization [36] and tourism development; meanwhile, a few scholars have also paid attention to the coupling degree between the inbound tourism

market and the environment [37] and export trade [38]. The research methods mostly include the coupling coordination model and coupling correlation model.

While scholars have conducted many studies on the spatial pattern of tourism development using social network analysis, spatial autocorrelation analysis, the coefficient of variation, and other research methods that look mainly at the national scale or urban agglomerations at the regional and provincial scales, relatively few studies have been conducted at the county scale or for individual cities. Hence, we focus on Beijing, a representative megacity in China, as the study area and use data on tourism development between 2010 and 2019 to study the spatial pattern of its tourism development, aiming to provide a reference for decision making on tourism development in China's megacities.

## 3. Study Area, Data and Methodology

### 3.1. Study Area

Beijing is located at 39.4° N–41.6° N 115.7° E–117.4° E and covers an area of 16,410 km$^2$ in total. The city comprises 16 districts, including Dongcheng, Xicheng, and Chaoyang. The topography is high in the northwest and low in the southeast, it includes notable mountains such as Yunmeng, Shangfang, and Mang, and rivers including Yongding, Chaobai, Beiyun, and Juma flow through the city (Figure 1). These abundant mountains and water resources provided prerequisites for tourism development in Beijing. China's capital is also a modern metropolis, famous as a historical and cultural city, with a history of over 3000 years and many imperial buildings, such as the Forbidden City, Summer Palace, and Yuanmingyuan (Old Summer Palace). These historical and cultural resources have led to the rapid development of the tourism industry in Beijing. Although the COVID-19 pandemic affected the tourism industry in 2020, Beijing's tourism revenue was still as high as CNY 291.4 billion, ranking first in urban tourism in China. Therefore, the development of Beijing's tourism industry is an important part of China's overall tourism development. Although the tourism industry in Beijing is well-developed, there are enormous differences among the areas of the city. The two main urban areas, Dongcheng and Xicheng, rely on the rich culture of imperial cities to attract tourists. Secondary urban districts, such as Chaoyang and Fengtai, rely on advanced modern atmospheres to attract large numbers of tourists. However, Mentougou, Huairou, and Miyun are far from the city center and attract fewer tourists, so the tourism industry in these areas is poorly developed. Thus, Beijing is representative of China's tourism industry, making it a suitable case study to investigate the spatial pattern of tourism development and influencing factors that can serve as a reference for developing tourism in Beijing and reducing regional differences, as well as providing theoretical support for policies related to treating big city disease.

### 3.2. Research Methodology and Data Sources

#### 3.2.1. Research Methodology

Coefficient of variation (CV). The coefficient of variation reflects the degree of dispersion in tourism development. Compared with other research methods, this coefficient has the advantage that it does not need to refer to the average value of the data and has a dimensionless quantity. Hence, we used it to measure the spatial variation in tourism development in Beijing. This is calculated as follows [39]:

$$CV = \frac{\sqrt{\frac{1}{n}\sum_{i=1}^{n}(x_i - \overline{x})^2}}{|\overline{x}|} \tag{1}$$

In Equation (1), CV is the coefficient of variation, $x_i$ denotes the low i variable value, $\overline{x}$ is the mean value of each variable, and n denotes the total number of variables. The larger the CV value of tourism development, the greater the density and sparsity of tourism development in Beijing and the more pronounced the regional differences.

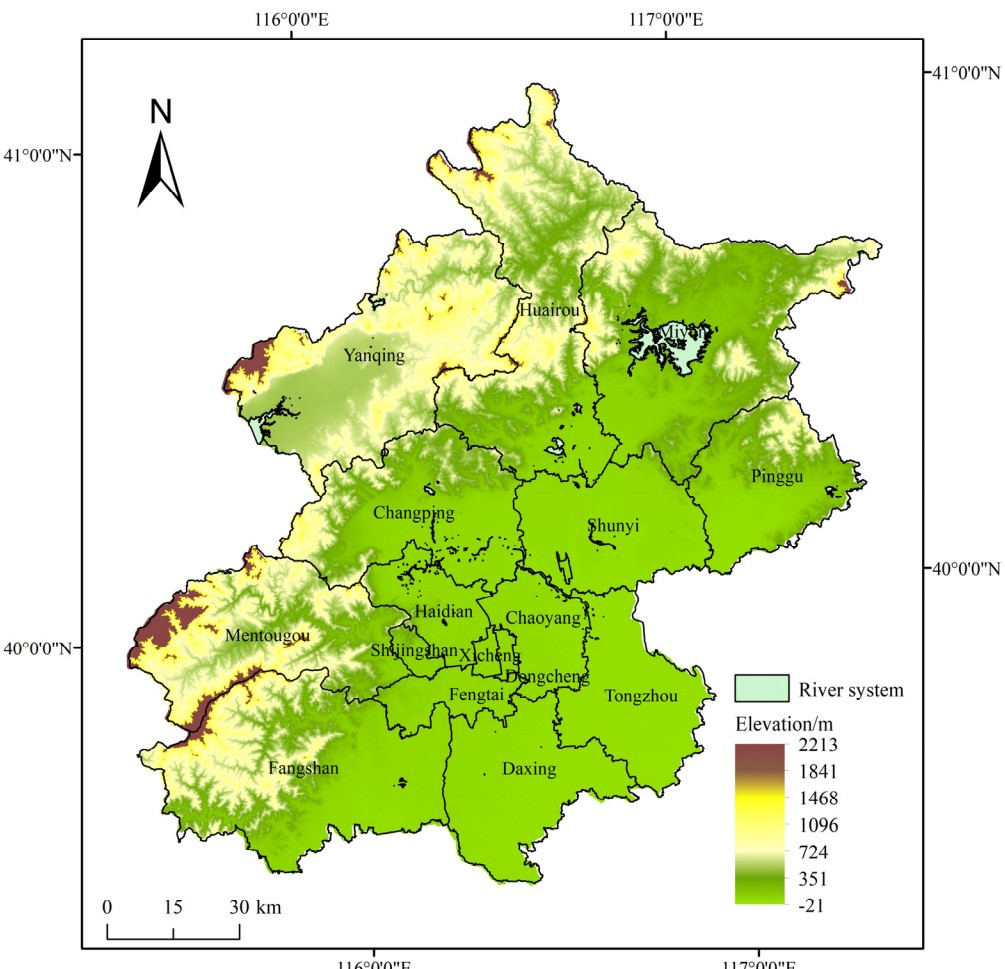

**Figure 1.** Regional division map of Beijing (drawn by the author).

Spatial autocorrelation. The spatial autocorrelation coefficient can provide insights into the spatial distribution pattern and agglomeration of tourism development. The global autocorrelation coefficient, expressed by Moran's I, was used to demonstrate the degree of spatial similarity among areas. The Moran's I index is one of the most widely used global indices. It can only reflect whether there are aggregation characteristics in the adjacent areas of the study area by virtue of a single attribute. Simultaneously, compared with other research methods, such as social network analysis and the modified gravity model, its data availability is strong, and the results are more intuitive. Local spatial autocorrelation was used to demonstrate the degree of association of local areas, as measured using local Moran's I. These values were calculated as follows [40]:

$$\text{Moran's I} = \frac{\sum_{i=1}^{n} \sum_{j=1}^{n} w_{ij}(x_i - \bar{x})(x_j - \bar{x})}{s^2 \sum_{i=1}^{n} \sum_{j=1}^{n} w_{ij}} \tag{2}$$

In Equation (2), I is the global spatial autocorrelation coefficient. The higher the value of I, the greater the spatial relevance of tourism development in Beijing. $x_i$ and $x_j$ are the observed values of the geographical attributes of tourism development. n is the sample size. $\bar{x}$ and $s^2$ are the mean and variance of observation s, respectively. $w_{ij}$ is the spatial weight matrix constructed based on the proximity criterion.

$$I_i = z_i \sum_{j=1}^{n} w_{ij} z_j \tag{3}$$

In Equation (3), $z_i$ and $z_j$ are the normalized values of the observations of areas i and j, respectively, and $I_i$ is the local spatial correlation of economic growth and industrial structure change between areas i and j, which can be classified into the following four cluster types: high–high, low–high, high–low, and low–low.

Geodetector. A geodetector is an important statistical method used to explore spatial analysis and reveal driving forces [41]. Compared with other traditional methods, this research method does not require explicit linear assumptions, has higher accuracy when the number of samples is <30, and can satisfy the interaction of detection factors [42]. It includes four main components: risk, factor, ecological, and interaction detection. Therefore, the geodetector method is more suitable for small-scale research because our sample size included 16 districts. We used this method to measure the explanatory power of the influencing factors affecting the spatial differentiation of tourism development in Beijing but with only two types of factor and interaction detection. These values were calculated as follows [41]:

$$q = 1 - \frac{\sum_{h=1}^{L} N_h \sigma_h^2}{N \sigma^2} \qquad (4)$$

Factor detection. In Equation (4), q is the explanatory power of the influence factor on the spatial pattern of tourism development in Beijing, $N_h$ is the number of sample units in the sub-region, n is the number of sample units in the whole region, L is the number of sub-regions, $\sigma^2$ is the variance of the sample size, $N_h \sigma_h^2$ is the variance of sub-regions, and the range of q is [0, 1]. The higher the value, the greater the influence of the factor on the spatial pattern of tourism development, and the lower the value, the less the spatial pattern of tourism development is driven by the influencing factor.

Interaction detection. The interaction detection works as follows [41]: if q (X1∩X2) = q (X1) + q (X2), then factors X1 and X2 are independent; if Min (q (X1), q (X2)) < q (X1 ∩ X2) < Max q (X1), q (X2)), then the single-factor nonlinearity after the interaction of factors X1 and X2 decreases; if q (X1∩X2) < Min (q (X1), q (X2)), then the nonlinearity decreases after the interaction of X1 and X2; if q (X1∩X2) > q (X1) + q (X2), then the nonlinearity increases after the interaction of X1 and X2; and if q (X1∩X2) > Max (q (X1), q (X2)) and q (X1∩X2) < q (X1) + q (X2), then X1 and X2 enhance each other after interaction.

### 3.2.2. Data Derived from Indicator Descriptions

Tourism revenue was used as an indicator of tourism development, following the literature. While data on the tourism revenue of each district in Beijing are available only up to 2010, we were able to compile data on the district-level tourism revenue from 2010 to 2019 using data from the *Beijing Statistical Yearbook*, *Beijing Districts Statistical Yearbook*, and the official website of the Beijing Municipal Bureau of Culture and Tourism.

## 4. Spatial Distribution Characteristics of Tourism Development in Beijing
### 4.1. Spatial Differences in Tourism Development
#### 4.1.1. Analysis of Coefficient of Variation

Based on Equation (1), we calculated the coefficient of variation of tourism development between 2010 and 2019 using the tourism revenue of each district in Beijing (Table 1). To explore in depth the current differences within Beijing, we divided the 16 districts of Beijing into core districts (Dongcheng and Xicheng), urban function expansion districts (Chaoyang, Haidian, Fengtai, and Shijingshan), new urban development districts (Tongzhou, Daxing, Shunyi, and Changping), and ecological conservation districts (Mentougou, Fangshan, Pinggu, Huairou, Miyun, and Yanqing), and calculated the coefficients of variation for each. The results are presented in Table 1.

**Table 1.** Variation coefficient of regional tourism development in Beijing.

|  | 2010 | 2011 | 2012 | 2013 | 2014 | 2015 | 2016 | 2017 | 2018 | 2019 |
|---|---|---|---|---|---|---|---|---|---|---|
| Overall districts | 1.195 | 1.201 | 1.201 | 1.202 | 1.207 | 1.214 | 1.217 | 1.226 | 1.237 | 1.217 |
| Core districts | 0.190 | 0.210 | 0.210 | 0.220 | 0.230 | 0.230 | 0.230 | 0.230 | 0.060 | 0.050 |
| Urban function expansion districts | 0.830 | 0.830 | 0.810 | 0.820 | 0.860 | 0.860 | 0.870 | 0.880 | 0.830 | 0.740 |
| New urban development districts | 0.480 | 0.470 | 0.470 | 0.450 | 0.430 | 0.410 | 0.390 | 0.390 | 0.280 | 0.280 |
| Ecological conservation districts | 0.337 | 0.309 | 0.305 | 0.293 | 0.287 | 0.316 | 0.321 | 0.333 | 0.379 | 0.420 |

The results of the overall regional coefficient of variation calculation show that: (1) the spatial variation of tourism development in Beijing increased from 1.195 in 2010 to 1.217 in 2019; and (2) it had a fluctuating upward trend, with a small increase in variation from 2010 to 2013, but the increase in variation in 2014 and subsequent years gradually increased. However, a decrease was observed in 2019.

The results of the subregional calculations are as follows: (1) We found spatial differences in tourism development among all districts; however, the degree of difference was smaller than the overall degree of difference. (2) The degree of variation in tourism development among the districts varied greatly, and the 10-year average coefficient of variation for the four district groups (Figure 1) showed the degree of variation: urban function expansion districts (0.833), new urban development districts (0.405), ecological conservation districts (0.330), and core districts (0.185). (3) The spatial differences in tourism development in the core, urban expansion, and ecological conservation districts first increased and then decreased, and only the spatial differences in tourism development in the new urban development districts decreased.

4.1.2. Local Variation in Tourist Visits

To clarify the spatial differences in tourism development in Beijing, we used ArcGIS 10.7 to map regional differences in tourism visits from 2010 to 2019 at 2-year intervals (Figure 2).

Figure 2 indicates the following: (1) Tourism visits in Beijing were concentrated mainly in Dongcheng, Xicheng, and Haidian, while Mentougou was the area with the lowest number of tourism visits. (2) During the study period, the number of tourists to each district also changed notably. For example, in Huairou, with the recent development of tourism, the number of B&Bs gradually increased, and tourism visits crossed from the second to the third stratum in 2011. With the acquisition of the right to host the 2022 Winter Olympics in China, a large number of ski resorts have been built in Yanqing, which has attracted many tourists, helping the district cross from the second to the third stratum. Furthermore, Fengtai recently invested in building tourism infrastructure. The number of travel agencies in the district increased from 74 in 2011 to 115 in 2013, and the number of tourists increased significantly as well, crossing from the third to the fourth stratum in 2011. (3) In terms of the division of urban development functions, tourism visits are concentrated in the core region, and the development pattern of tourism is core districts, urban function expansion districts, ecological conservation districts, and new urban development districts.

*4.2. The Spatial Relevance of Tourism Development*

4.2.1. Global Autocorrelation Analysis

This study found a significant spatial correlation in tourism development in Beijing (Table 2). Furthermore, we calculated Moran's I for tourism development from 2010 to 2019 using a geographic adjacency matrix and Equation (2). The results show a spatial correlation in tourism development in Beijing during this period, significant at the 1% level. Hence, tourism development in Beijing is significantly correlated, and tourism development between areas is not random, but is influenced by other surrounding areas.

**Table 2.** Spatial autocorrelation coefficient of tourism development in Beijing.

| Year | 2010 | 2011 | 2012 | 2013 | 2014 | 2015 | 2016 | 2017 | 2018 | 2019 |
|------|------|------|------|------|------|------|------|------|------|------|
| Date | 0.375 | 0.381 | 0.398 | 0.399 | 0.397 | 0.387 | 0.391 | 0.388 | 0.413 | 0.426 |

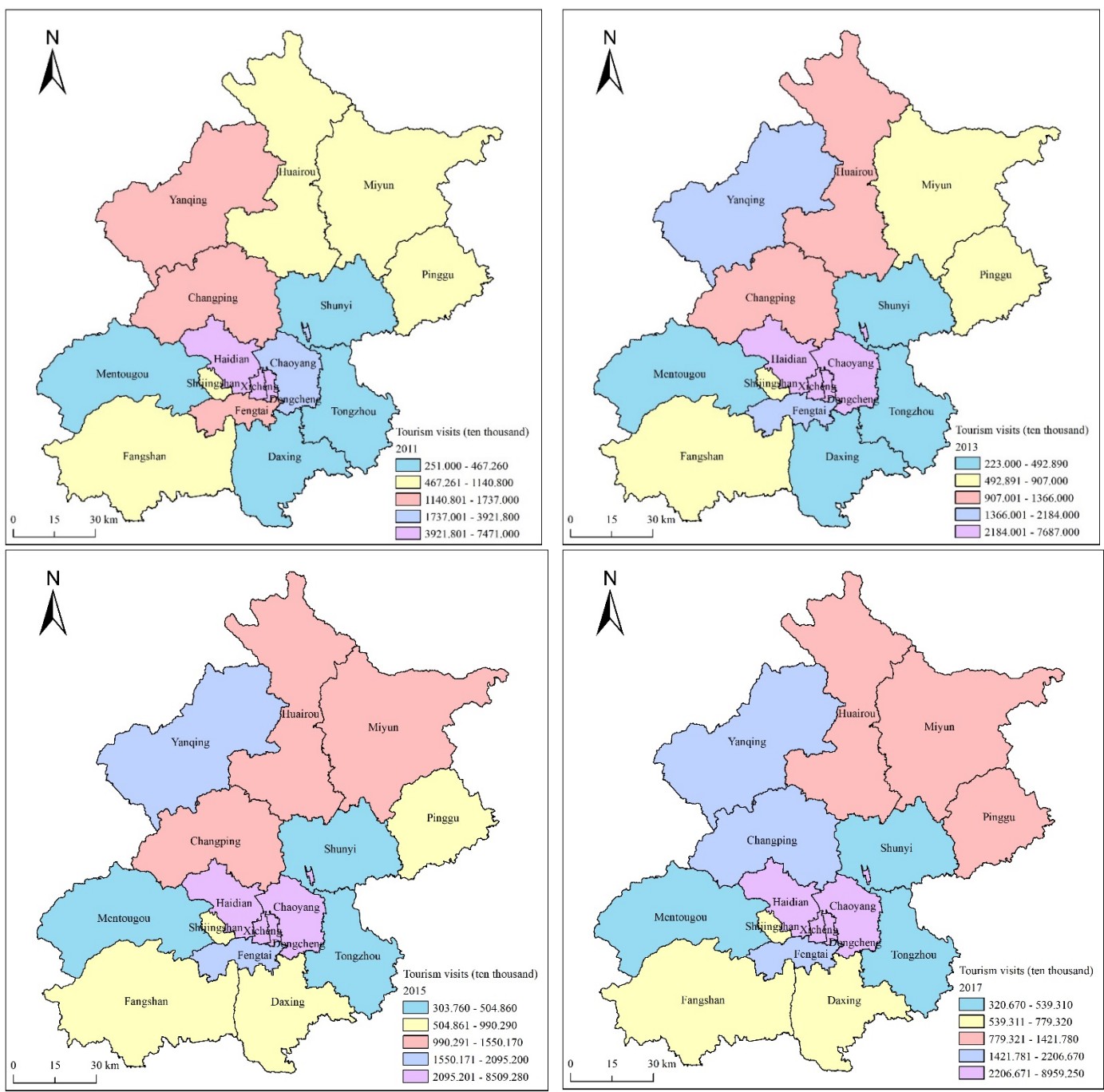

**Figure 2.** *Cont*.

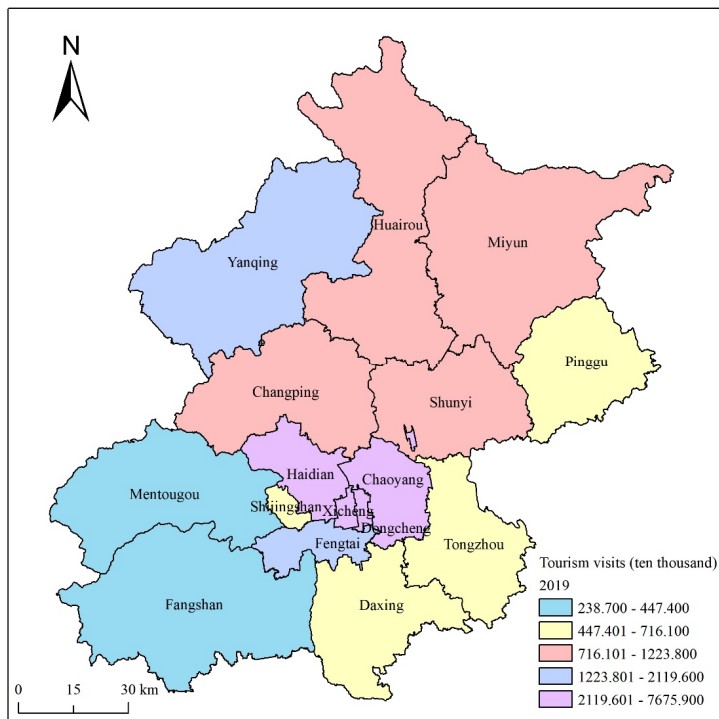

**Figure 2.** Regional differences in tourism visits in Beijing in 2011, 2013, 2015, 2017, and 2019 (drawn by the author).

### 4.2.2. Local Indicators of Spatial Association Analysis of Spatial Local Correlations

Because Moran's I has great limitations in testing spatial autocorrelations, it cannot detect the local spatial correlation and heterogeneity in each direction. Therefore, we require a local index that reflects spatial correlation to determine whether there is a local spatial agglomeration of observations to portray the local spatial interdependence and spatial heterogeneity characteristics of industrial structural upgrading. We use Equation (3) to calculate the spatial local correlations state of Beijing and use LISA (local indicators of spatial association) cluster maps of tourism revenue in Beijing for 2011, 2013, 2015, 2017, and 2019 (Figure 3), with 2 years as the time node. All results are significant at the 1% level.

Figure 3 shows that during the study period, (1) the core districts and the urban function expansion districts of Beijing are basically in a high–high cluster state, the new urban development districts are basically in a high–low cluster state, and the ecological conservation districts are mostly in a low–low cluster state. The core districts and urban function expansion districts of Beijing are rich in tourism resources and infrastructure. They are also the gathering place of many 5A scenic spots, so they have attracted many tourists and created higher tourism revenue. While the new urban development districts are close to the core districts and have relatively perfect infrastructure, the ecological conservation districts are located in the periphery of Beijing, mostly in mountainous areas, with less road construction, fewer travel agencies and star-rated hotels, and a lack of sufficient publicity. Therefore, they are not attractive to tourists, and their tourism revenue is relatively small. (2) The spatial pattern of tourism development in Beijing is basically stable, but in 2015 and 2017, Chaoyang district changed from the original high–high cluster state to a high–low cluster state, and Shunyi district changed from a low–low cluster state to a low–high cluster state. The main reason for this phenomenon is that under the new tourism planning, the road construction and ecological construction in Shunyi district have promoted the rapid development of business tourism and leisure tourism, and the tourism revenue is higher than that of the surrounding areas.

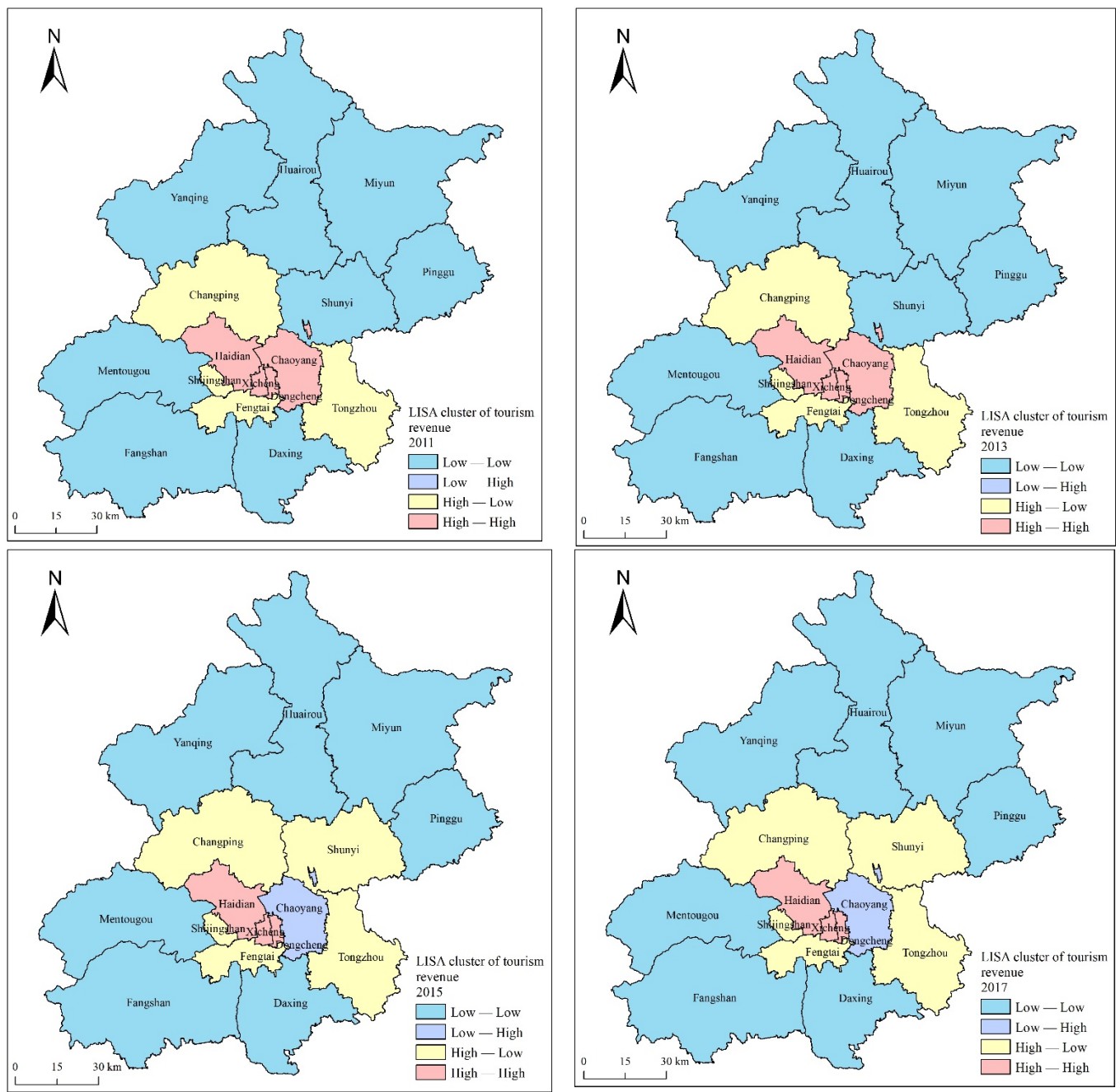

**Figure 3.** *Cont.*

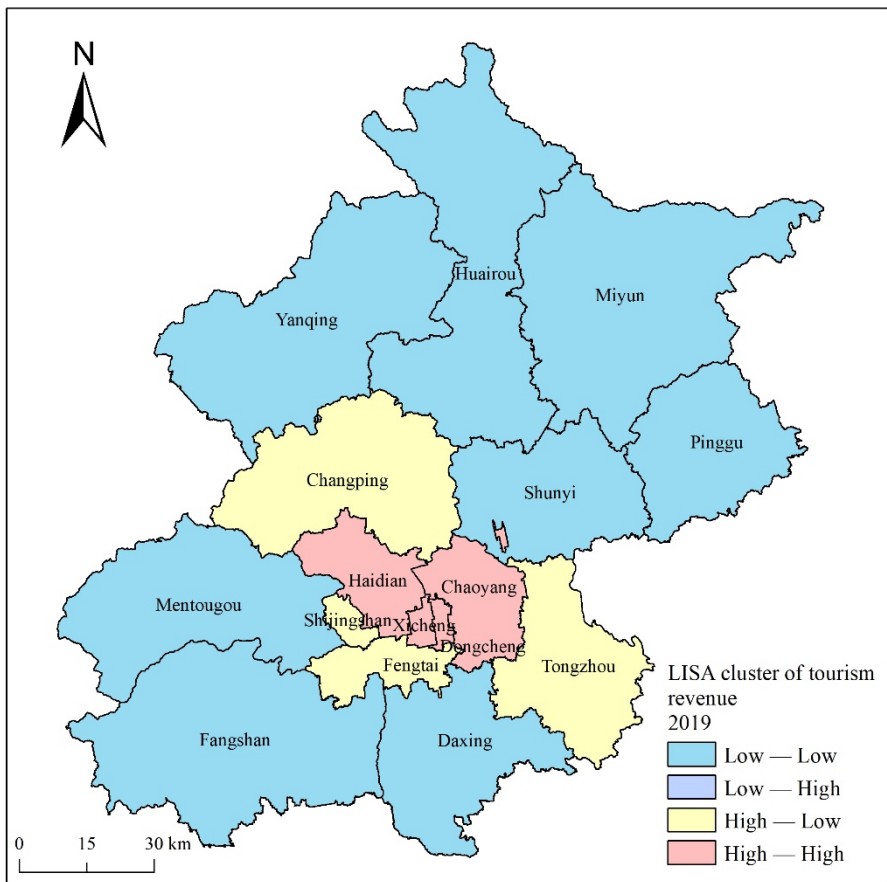

**Figure 3.** Local LISA cluster map of tourism revenue in Beijing (drawn by the author).

In conclusion, our analysis indicates significant spatial autocorrelation and spatial dependence of tourism development in Beijing; that is, tourism development in one area drives development in neighboring areas.

## 5. Exploring the Influence Mechanism of Tourism Development in Beijing

### 5.1. Identifying Influencing Factors

Several factors can influence spatial patterns of tourism development. Based on the literature, we identified the tourism factor inputs, tourism reception facilities, and urban development levels (Table 3). Additionally, we processed the data and classified them into five levels using the natural breakpoint method [43].

Tourism factor inputs: Tourism factor inputs mainly include tourism resources and labor. Tourism resources are the material conditions for tourism development and the carrier and basis for all tourism activities. To ensure the availability of data and the authority of our research, we followed the method by Zhou [44]. Based on actual circumstances, we used the number of regional 5A and 4A scenic spots to perform weighting calculations to obtain the tourism resource endowment of individual areas. The tourism resource endowment is $n_1 \times 4 + n_2 \times 5$, where $n_1$ and $n_2$ indicate the number of 4A and 5A scenic spots, respectively. Tourism development is linked to labor input, which is one of the most important factors affecting tourism development. Because official statistics do not provide the number of tourism employees, scholars have mostly used the number of employees in the tertiary industry as a substitute. Although this indicator magnifies the scale of labor input factors in tourism, it fully reflects the comprehensive characteristics of urban tourism. We followed previous research [45] and used the number of tertiary-sector employees.

**Table 3.** Influencing factors, indicators, and sources.

| Influence Factor | Indicator Source | Index Measure | Data Sources |
|---|---|---|---|
| Tourism factor inputs | Tourism resources (X1) | Tourism resource endowment | The official website of the Beijing Municipal Bureau of Culture and Tourism |
| | Tourism labor (X2) | The number of tertiary sector employees. | Beijing Districts Statistical Yearbook |
| Tourism reception facilities | Tourism infrastructure (X3, X4) | The number of star-rated hotels The number of travel agencies | The official website of the Beijing Municipal Bureau of Culture and Tourism |
| Urban development level | Level of economic development (X5) | GRP | Beijing Districts Statistical Yearbook |
| | Level of urbanization (X6) | The urban population/the total resident population | |
| | Degree of openness (X7) | The amount of actually utilized foreign capital | |

Tourism reception facilities: Tourism activities include food, accommodation, transportation, shopping, and entertainment. Tourism reception facilities can promote the development of tourism activities and play a significant role in local tourism development. We use the number of star-rated hotels and travel agencies to determine the level of tourism reception facilities in an area.

Level of urban development: The level of urban development includes the level of economic development, level of urbanization, and degree of openness. The economic development level of a city provides capital support and material guarantees for the development of urban tourism, increases residents' tourism consumption, and expands the tourism market. We used gross regional product (GRP) to express the economic development level of Beijing. The process of urbanization involves the spatial agglomeration of people and industries, which can promote the expansion of tourism and tourism consumption, as well as the optimization of the tourism industry structure, clustering of tourism factors of production, and tourism innovation spillovers. We used the ratio of the urban population to the total resident population to represent the urbanization level. Increasing a city's openness to the outside world is conducive to improving the capital and technology levels of the tourism industry, and increases the number of inbound businesses and other types of tourists. We adopted the amount of utilized foreign capital to represent Beijing's openness to the outside world.

*5.2. Results and Analysis*

5.2.1. Factor Detection Results

Based on Equation (4), we calculated the detection q values for each impact factor X1–X7 using the geodetector. The results are presented in Table 4.

**Table 4.** Impact factor geographic detection results.

| | X1 | X2 | X3 | X4 | X5 | X6 | X7 |
|---|---|---|---|---|---|---|---|
| 2010 | 0.248 | 0.636 ** | 0.969 *** | 0.916 *** | 0.714 ** | 0.555 * | 0.661 |
| 2011 | 0.414 | 0.811 ** | 0.892 *** | 0.871 ** | 0.693 ** | 0.550 * | 0.794 * |
| 2012 | 0.514 | 0.806 ** | 0.959 *** | 0.870 *** | 0.639 * | 0.558 * | 0.650 |
| 2013 | 0.331 | 0.806 ** | 0.959 *** | 0.875 *** | 0.688 ** | 0.555 * | 0.801 * |
| 2014 | 0.431 | 0.790 ** | 0.962 *** | 0.887 *** | 0.667 * | 0.532 | 0.233 |
| 2015 | 0.482 | 0.798 ** | 0.966 *** | 0.909 *** | 0.635 * | 0.528 | 0.629 |
| 2016 | 0.465 | 0.788 ** | 0.857 *** | 0.897 *** | 0.627 * | 0.520 | 0.626 |
| 2017 | 0.455 | 0.788 ** | 0.854 *** | 0.899 *** | 0.629 * | 0.515 | 0.764 |
| 2018 | 0.513 | 0.786 ** | 0.968 *** | 0.862 *** | 0.761 ** | 0.513 * | 0.543 |
| 2019 | 0.415 | 0.823 *** | 0.963 *** | 0.692 | 0.756 ** | 0.614 * | 0.476 |

Note: *** indicates significance at the 1% level, $p < 0.01$. ** indicates significance at the level of 5%, $0.01 < p < 0.05$. * indicates significance at the 10% level, $0.05 < p < 0.1$.

The q-values of the seven factors indicate that most have significant effects on the spatial pattern of tourism development in Beijing. Star-rated hotels and travel agencies have the greatest effect, and the number of employees in the tertiary industry and GRP have the second greatest effect. The endowment of tourism resources does not affect the spatial pattern of tourism development in Beijing, which is influenced only by the level of urbanization and the degree of openness in certain years. The impact of the seven factors on the spatial pattern of Beijing's tourism development is as follows.

Tourism resource endowment (X1): from 2010 to 2019, the effect of tourism resource endowment on the spatial pattern of tourism development in Beijing gradually increased, but did not pass the 10% level of significance. Many scholars consider tourism resource endowment as the original driving force of tourism development [46,47]. This has a significant impact on the development of tourism in the area. However, Beijing has many tourism resources, and the differences between the districts are not very obvious. In addition, just eight 5A scenic spots, including the Forbidden City and the Great Wall, attract approximately 10% of Beijing's annual tourist traffic, resulting in a strong concentration of tourists. Moreover, we did not include scenic spots rated 3A and below in tourism resource endowment, which ignores the impact of such scenic spots on the spatial pattern of tourism development. Consequently, tourism resource endowment does not have a significant impact on the spatial pattern of tourism development in Beijing.

Tertiary sector employees (X2): The number of tertiary sector employees has a gradually increasing influence on the spatial pattern of tourism development in Beijing. Tourism is a labor-intensive industry that requires labor input. Tourism is an important part of the tertiary industry; therefore, the number of employees in this sector has an important positive effect on the spatial pattern of tourism development.

Tourism reception facilities (X3, X4): Tourism reception facilities are the dominant factor influencing tourism development in Beijing. Both star-rated hotels and travel agencies contributed significantly to the spatial pattern of tourism development, but the extent of their contribution fluctuated during the study period. Between 2010 and 2015, star-rated hotels and travel agencies were located mainly in Chaoyang, Dongcheng, Xicheng, and Haidian, with large differences between districts. However, with the continuous development of tourism, the differences gradually narrowed, so the influence on the spatial pattern of tourism development slowly weakened. A recent policy aimed at decentralizing Beijing's non-capital functions provided numerous development opportunities in Tongzhou, Daxing, and Shunyi, and the star-rated hotels and travel agencies that sprung up in these areas. This further increased the differences between the districts. The construction of tourism reception facilities attracted many tourists [48,49], boosting local tourism revenues; thus, the extent of this factor's influence on the spatial pattern of tourism development is again increasing.

Gross regional product (X5): The impact of GRP on the spatial pattern of tourism development in Beijing is cyclical, with a gradual decrease from 2010 to 2017 and a jump in 2018, but then a further decrease. The GRP represents the overall economic condition of a region, and economically developed regions have a better industrial structure, more developed infrastructure, and good service quality for tourists, thus leading to a higher level of local tourism development. Thus, most scholars have argued that GRP is an important factor that influences regional tourism development [50,51]. However, some areas of Beijing, such as Chaoyang and Haidian, have significant siphon effects that cause fluctuations in the influence of GRP on tourism development. During 2010–2017, areas with a higher GRP in Beijing remained the main urban and suburban areas, such as Chaoyang and Haidian. Recently, however, national policies on decentralizing non-capital functions in Beijing and synergistic development in the Beijing–Tianjin–Hebei region have led some industries and enterprises to move. This policy promoted development in Daxing and Tongzhou, resulting in changes in GRP and the creation of a new spatial pattern. Thus, GRP has a cyclical impact on the spatial pattern of tourism development.

Level of urbanization (X6): The level of urbanization had a stable promotional effect on tourism development during the period, but with variations between years. When the level of urbanization increased, population and industry clustering also increased, and investment in tourism and tourism consumption increased, indicating a promotional effect on tourism development [52,53]. However, once the degree of urbanization development solidified, the promotional effect on tourism development was no longer visible. Urbanization in Beijing's districts increased between 2010 and 2013 but was not significant between 2013 and 2019. Influenced by development in Beijing, the urbanization of Tongzhou, Daxing, and Yanqing is gradually accelerating; therefore, urbanization has re-emerged as an important factor affecting tourism development.

Degree of openness (X7): During the study period, the degree of openness contributed to the spatial pattern of tourism development in 2011 and 2013. As an international metropolis, Beijing's degree of openness is evident, but foreign direct investment has a relatively small impact on its economic output; thus, the impact of foreign investment is less likely to appear in tourism development. Although some scholars have found that increasing openness can disrupt the ecological balance of an area to some extent [54] and reduce economic efficiency [55], reducing economic efficiency is still an important way to increase regional inbound tourism arrivals and foreign exchange earnings [56].

### 5.2.2. Interaction Detection Results

We explored whether the effect of the two factors on the spatial pattern of tourism development in Beijing is strengthened or weakened using the 10-year mean values of the six factors that passed the statistical test for interaction detection [57]. The detection results are listed in Table 5.

**Table 5.** Interaction factor detection results.

| Reciprocal Factor | q | Type | Reciprocal Factor | q | Type |
|---|---|---|---|---|---|
| X2∩X3 | 0.989 | BE | X3∩X7 | 0.997 | BE |
| X2∩X4 | 0.996 | BE | X4∩X5 | 0.839 | BE |
| X2∩X5 | 0.997 | BE | X4∩X6 | 0.827 | BE |
| X2∩X6 | 0.989 | BE | X4∩X7 | 0.946 | BE |
| X2∩X7 | 0.998 | BE | X5∩X6 | 0.834 | BE |
| X3∩X4 | 0.879 | BE | X5∩X7 | 0.790 | BE |
| X3∩X5 | 0.890 | BE | X6∩X7 | 0.940 | BE |
| X3∩X6 | 0.879 | BE | | | |

Note: BE (BI factor enhancement) refers to double factor enhancement.

The degree of interaction between factors affecting the spatial pattern of tourism development is significantly enhanced and mainly shows two-factor enhancement, indicating that no single influencing factor caused the difference in the spatial pattern of tourism development in Beijing. Instead, the difference was the result of the joint action of multiple influencing factors. The specific detection results are as follows.

The q-values of the eight interaction detection results were >0.9; six were >0.8, and one was <0.8. The degree of influence after the interaction of various factors was greater than that of the individual factors. Thus, tourism development is the result of the joint action of multiple factors; therefore, it is also influenced by the interactions of multiple factors.

The interaction between the number of star-rated hotels and degree of openness had the largest q-value, indicating that their joint effect was the greatest factor in the spatial pattern of tourism development in Beijing. Star-rated hotels, as the factor with the highest degree of influence among individual factors, are important for tourism development. Deepening the degree of openness increases the number of inbound tourists, further affecting tourism development. Thus, the interaction between star-rated hotels and openness is an important factor that affects tourism development in Beijing.

The interaction between GRP and openness had the lowest q value, indicating that their joint effect had the least influence on the spatial pattern of tourism development in

Beijing. The influence of GRP on tourism development is mostly cyclical, and GRP also influences the degree of openness to an extent. Therefore, the interaction between the two does not significantly boost tourism.

## 6. Discussion

Based on the two perspectives of spatial correlation and spatial analysis, this paper takes Beijing as a study, explores the spatial pattern of tourism development in China's megacity, and deeply analyzes the impact factors such as tourism investment, tourism infrastructure and urban development level on the spatial pattern of tourism development. Compared with the previous studies of scholars, this study is innovative and different from the conclusions of previous scholars, but it also has some limitations.

### 6.1. Spatial Form of Tourism Development at the Micro Scale

This study takes 16 regions of Beijing as the research samples, which is different from the national [5,10], urban agglomeration [12,14] and provincial [6] research samples commonly used by scholars before, and is more microscopic. As an important part of urban agglomerations, provincial units and the nation, cities are an important carrier of China's urbanization development. At the same time, cities, as the main space for the agglomeration of economic activities, are of great significance to the development of the country. Tourism is an important part of urban functions, and its development can cause the agglomeration of resources, industries and economies to a certain extent. Tourism has become an important pillar of the service industry and a new economic growth point, which plays a huge role in promoting urban economic development, labor employment, urban image, industrial structure optimization, international exchanges and cooperation, the improvement in the quality of life for citizens, and the coordinated development of environment, society and economy. Therefore, it is of great significance to study the spatial pattern of intercity tourism development.

### 6.2. The Spatial Pattern of Tourism Development Is an Important Factor to Promote the Change in Urban Spatial Pattern

With the rapid development of China's economy, the population and industries continue to gather in developed cities, forming a new urban scale, the "super city". How to carry a large population and industrial scale in a limited space requires a reasonable spatial layout. As an important part of urban functions, the spatial pattern of tourism development promotes a certain level of regional spatial coordination and cooperation, and the balanced growth of developed and underdeveloped areas, and is an important factor in adjusting the balance and coordinating development between areas. From our research, we find that areas with good tourism development are generally areas with high levels of population agglomeration, industrial agglomeration and economic development in cities. Thus, the spatial pattern of tourism development plays a pivotal role in the spatial governance and coordinated development of megacities. Therefore, changing the spatial pattern of tourism development can alleviate the problem of "big city disease" to a certain extent. For example, new tourism growth poles can be cultivated in regions with an insignificant population and industrial agglomeration, so as to promote the investment of capital and infrastructure, attract more people and industries from regions with significant agglomeration, and thus promote the transformation of urban spatial form. In regions with low levels of economic development, the construction of tourism infrastructure and node tourism attractions can be increased, and the radiation and driving role of regions with high levels of economic development expanded, so as to alleviate the development imbalance between urban regions.

### 6.3. Tourism Resource Endowment Is No Longer an Important Factor Affecting the Development of Urban Tourism

Many scholars have found that tourism resource endowment plays an important role in the development of tourism [46], but this study has not found the same conclusion.

Tourism resources are an important factor to attract tourists, but for urban tourism, it is not a core competitive advantage, and areas with more tourism reception facilities can increase the overnight rate of tourists, thereby increasing local tourism revenue. Especially in recent years, most tourists are affected by the Internet and have strong curiosity and hunting psychology; however, most products in traditional scenic spots have a simple connotation, single form, and are mostly landscape ornamental products, so they are not very attractive to tourists. On the contrary, the emerging night economy and online celebrity economy tourism resources are highly attractive to tourists and have a great impact on traditional scenic spots. Therefore, when improving the current situation of tourism development, regions can pay more attention to the construction of tourism infrastructure and the development of new business forms, and change the marketing methods of traditional scenic spots to promote their innovative development.

*6.4. Research Limitations*

This study has several limitations. Firstly, the spatial pattern of tourism development involves many fields. However, we only analyze the spatial differences and relevance of tourism development, while ignoring the spatial spillover and coordination of tourism development. Secondly, there are many factors that affect the spatial pattern of tourism development. At the same time, due to the limited available data, we only study the three dimensions of tourism factor investment, hotel facilities and urban development level, excluding the accessibility of tourism destinations, the role of government and so on. Therefore, the number of influencing factors can be increased in subsequent research. Thirdly, there are also limitations in using foreign investment to express the degree of openness, because it cannot accurately reflect the degree of cultural openness of a region, and thus the openness measurement can be further improved. Therefore, in future research, scholars need to pay attention to other spatial forms of tourism development, look for new measurement methods for the level of tourism openness, and increase the impact of other factors such as destination traffic accessibility and the level of government intervention.

**7. Conclusions**

Based on tourism development data in Beijing for 2010–2019, we applied the coefficient of variation, global autocorrelation, and local autocorrelation to analyze the spatial pattern of tourism development in terms of spatial differences and spatial correlations. We also used a geodetector to analyze the spatial pattern of tourism development from the three dimensions of tourism factor inputs, tourism reception facilities, and urban development level. What follows are the main conclusions.

In terms of spatial differences, the overall regional differences in tourism development in Beijing have been expanding, and the differences among the four regions first increase and then decrease. The internal differences among areas can be categorized as follows: urban function expansion districts, new urban development districts, ecological conservation districts, and core districts. Dongcheng, Xicheng, Chaoyang, and Haidian were the main areas for tourist arrivals, and the overall spatial pattern of tourism visits was core districts, urban function expansion districts, ecological conservation districts, and new urban development districts. The policy implication of this study is that narrowing the differences in tourism development in urban functional expansion districts means narrowing the differences in regional tourism development. Cities may have certain disparities in tourism development depending on regional functions, but regions with the same functions should not have huge development differences. Therefore, narrowing the differences between small areas to drive coordinated development between large regions is a necessary means to promote the spatial distribution of tourism development.

In terms of spatial correlation, the overall correlation of tourism development in Beijing is gradually strengthening, and the degree of mutual influence between areas is increasing. Local spatial autocorrelation mainly consists of low–low clusters, followed by high–low clusters. The implication is to deepen the connections between these areas. Although each

factor influences tourism development, the process of tourism development is complex, and the flow of factors of production between areas and collaboration among areas will affect the development of other areas to some extent. Thus, deepening connections between areas is an important way to promote the spatial pattern distribution of tourism development in Beijing.

In terms of influencing factors, the single-factor detection results show that tourism factor inputs, tourism reception facilities, and urban development levels have different degrees of influence on the spatial pattern of tourism development in Beijing. The dominant influencing factor was tourism reception facility. Regarding tourism factor inputs, tourism resource endowment does not have a significant influence. The two-factor interaction detection results show that the spatial pattern of tourism development in Beijing is a result of the joint action of multiple factors rather than a single factor, among which the interaction between star-rated hotels and the degree of openness has a more drastic impact. Thus, expanding the construction of tourism reception facilities, especially star-rated hotels, and the level of openness is the most important factor in promoting the rational distribution of tourism development.

**Author Contributions:** Y.C. was the major writer of the manuscript; J.L. conceived of the idea, calculated some data, led the project, and acquired funding support. All authors read the first draft, helped with the revision, and approved the article. All authors have read and agreed to the published version of the manuscript.

**Funding:** The publication of the present work is supported by the National Natural Science Foundation of China (grant no. 41771131), Key Projects of Beijing Social Science Foundation (grant no. 21JCB050), and the Premium Funding Project for Academic Human Resources Development in Beijing Union University (grant no. BPHR2020AS02).

**Institutional Review Board Statement:** Not applicable.

**Informed Consent Statement:** Not applicable.

**Data Availability Statement:** The primary data used to support the findings of this study have been explained clearly.

**Conflicts of Interest:** The authors declare that there are no conflict of interest.

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
