# Peer review of "The Spatial Spillover Effect and Its Impact on Tourism Development in a Megacity in China"

_sustainability, doi:10.3390/su14159188_

Round 1

Reviewer 1 Report

Dear authors,

First of all, congratulations for your work. However, manuscript should be modified following considerations described as follows:

Line 2. Shorten title up to 12 words.

Lines 31-48. Please reinforce first two paragraphs with some references.

Line 87. Manuscript lacks of Literature Review, please add section and reinforce the study with some new (and current) references to the topic.

 Line 115. Please add source to Fig. 1 and add it to references.

Line 118. Add acronym and use it always beyond first time appeared: Coefficient of variation (CV). The CV reflects the degree…

Lines 118-169. You must cite equations in text body always you refer to: (1), (2), (3) and (4).

Line 174. Cite source correctly and add it to references.

Lines 194-203. Please do not enumerate results as can be confused as it has same coding as equations (1) and (2).

Line 209 and 241. Please refers always the same (Fig. or Figure) but never both versions.

Lines 209-223. Same as lines 194-203.

Line 234. Please add complete version of LISA acronym. Do not use acronyms in titles and subtitles.

Lines 244-256. Same as lines 194-203.

Lines 257-259. Shift to conclusions.

Line 265. Cite correctly sources in Table 3 then add them to references.

Line 270. Zhou et al. is not cited correctly.

Lines 304-305. You cannot write a paragraph of two lines. Please merge this paragraph with the second paragraph (lines 310-316) and shift just before Table 4 appearance.

Lines 413-430. In discussion section you have to compare your results with those obtained in the previous works cited in your literature review section.

Lines 431-445. Shift limitations to the last part of conclusions, just before the future lines of research. Remove numeration (1), (2), (3).

 Line 447. Please do not use first person, used impersonal style instead text along.

 Line 488. Add, in that order, practical applications, limitations and futures lines of research.

Author Response

Dear reviewer and editor:

Thank you very much for reviewing our paper, for the comments provided by the reviewers, and for the hard work of the editing teachers. These suggestions have important guiding significance for our paper writing and scientific research work! On the basis of carefully reading the review comments, we have made the following modifications:

Q1:Shorten title up to 12 words.

We accept your suggestion to shorten the title to 12 characters and change it to “Spatial Spillover Effect and Impact Factors of Tourism Development in Megacity China”. However, as the main method to explore the spatial pattern differences of tourism development in this paper, geographical detectors can be placed in the title of the article to make it more convenient for readers to refer to such research methods and screen this article. 

Q2: Please reinforce first two paragraphs with some references.

On the basis of carefully reading the relevant literature, we added some literature to support our views. For your convenience, we have temporarily marked the document yellow in the text.

  1. You, H. Y.; Yang, X. F. Urban Expansion in 30 Megacities of China: Categorizing the Driving Force Profiles to Inform the Urbanization Policy. Land Use Policy, 2017, 68, 531-551.
  2. Shen, J.; Zhang, K. Y. An Empirical Analysis of Factors Leading to Typical Urban Problems in China. Progress in Geography. China , 2020, 39, 1-
  3. Zhu, H. Y. The Relationship between Tourism Development and Economic Growth Based on the Threshold of City Scale. Tourism Science, China,2021, 35, 17-
  4. Wang, J. Y.; Zhang, H. Tourism Development, Spatial Spillover and Regional Development Imbalance. Tourism Science, China 2021, 35, 73-

Q3: Manuscript lacks of Literature Review, please add section and reinforce the study with some new (and current) references to the topic.

We are carefully reading the research literature at home and abroad, and believe that the research on the spatial pattern of tourism development mainly focuses on its spatial correlation, heterogeneity, spillover and spatial coupling. On the basis of the original, we have added a research review on the spatial coupling of tourism development, and described it in segments. And the references are replaced and added. For your convenience, we have temporarily marked the document yellow in the text.

  1. Wang, Y. W.; Xi, M. M.; Chen, H.; Lu, C. Evolution and Driving Mechanism of Tourism Flow Networks in the Yangtze River Delta Urban Agglomeration Based on Social Network Analysis and Geographic Information System: A Double-Network Perspective. Sustainability.2022, 14, 7656.
  2. Liu, C. L.; Qin, Y. J.; Wang, Y. F.; Yu, Y.; Liu, G. H. Spatio-Temporal Distribution of Tourism Flows and Network Analysis of Traditional Villages in Western Hunan. 2022, 14, 7943.
  3. Zhang, Z. Q.; Li, L.; Guo, Q. Y. The Interactive Relationships between the Tourism-Transportation-Ecological Environment System of Provinces along the ‘Silk Road Economic Belt’ in China. Sustainability.2022, 14, 3050.
  4. Feng, J. H.; Lin, Y.; Jiang, K. H.; Li, P. L.; Ye, G. Q. Spatiotemporal Coupling Coordination Measurement on Islands’Economy- Environment- Tourism System. Ocean & Coastal Management. 2021, 212, 105793.
  5. Weng, G. M.; Tang, Y. B.; Pan, Y.; Mao, Y. Q. Spatiotemporal Evolution and Spatial Difference of Tourism- Economy- Urbanization Coupling Coordination in Beijing- Tianjin- Hebei Urban Agglomeration. China 2021, 41, 196–204.
  6. Wang, J.; Zhang, Q. Y.; Li, X. S. Spatial Coupling and Evolution of Border Tourism System: A Case Study of Ruili City on the China- Myanmar Border. Areal Research and Development. China 2021, 40, 86–
  7. Bao, F. H.; Chen, Y. Evolution Characteristics and Driving Mechanism of Spatiotemporal Coupling of Gravity Centers between Inbound Tourism and Import- Export Trade in China. Tourism Tribune. China2019, 34, 66–

Q4: Please add source to Fig. 1 and add it to references.

Fig 1, 2 and 3 are drawn by ourselves, not for reference of other scholars. We also marked this in the text. For your convenience, we marked it in yellow.

Q5: Add acronym and use it always beyond first time appeared: Coefficient of variation (CV). The CV reflects the degree…

We have added the abbreviation of coefficient of variation and marked it in yellow.

Q6: You must cite equations in text body always you refer to: (1), (2), (3) and (4).

Thank you very much for your reminder. We did have an oversight. We have added references to equations (1), (2), (3) and (4) in the text, and marked them in yellow for convenience.

  1. Mu, X. Q.; Guo, X. Y.; Ming, Q. Z. The Spatio-Temporal Evolution and Impact Mechanism of County Tourism Poverty Alleviation Efficiency from the Perspective of Multidimensional Poverty: A Case Study of 25 Border Conunties (Cities) in Yunnan Province.Economic Geography. China 2020, 40, 199–
  2. Lesage, J.; Pace, R. K. Introduction to Spatial Econometrics. 2009.

Q7: In discussion section you have to compare your results with those obtained in the previous works cited in your literature review section.

In previous studies, scholars have found that there are different spatial patterns in tourism development, that is, conical, elliptical, triangular and other spatial correlation networks; Tourism development in different regions has great heterogeneity. In this study, we also found the existence of this phenomenon, but different from other scholars' research findings, the spatial form of tourism development is similar to the spatial form of urban development, which provides a new solution to the governance of "big city disease" in megacities, that is, from the spatial form of tourism development. For your convenience, we marked this part in yellow.

Q8: Shift limitations to the last part of conclusions, just before the future lines of research. Remove numeration (1), (2), (3).

After we mentioned the limitations of the article to the conclusion of the article, we deleted (1), (2), (3) and added future research directions. At the same time, for your convenience, we marked this part in yellow.

Once again, thank the reviewers and the editor for your valuable comments on this article. If there are suggestions for further revision, we will do our best to make the revision.

Best wishes .

Yours

Cao Yajun, Liu Jianguo

Reviewer 2 Report

Article: Spatial Spillover Effect and Impact Factors of Tourism Devel-2 opment in Megacity China Based on Geographic Detector 3 Method addresses the important issue of the role of tourism in the function of large cities. However, the choice of Beijing is hardly representative of other cities, not only in China but also in the world. Very poor data are analysed, however, the authors point out that there is no access to other statistical data. The analysis at the level of districts also seems to be very generalised.

In addition, I have noticed the following problems:

- if the analyzed data are revenues from tourism, should they be converted into the size of the district or as a share in total revenues?

- Fig. 2. cartogram maps are used for relative values, diagram maps are used for absolute values. No units of value.

- lack of information on the tourist infrastructure in the districts.

- it is only in chapter 4 that the tourist potential of the districts is analyzed

However, if the article is considered theoretical, presenting the application of selected research methods, it can be published after correction of the indicated problems.

Author Response

Dear reviewer and editor,

Thank you very much for reviewing our paper, for the comments provided by the reviewers, and for the hard work of the editing teachers. These suggestions have important guiding significance for our paper! On the basis of carefully reading the review comments, we have made the following modifications:

Q1: If the analyzed data are revenues from tourism, should they be converted into the size of the district or as a share in total revenues?

We believe that: (1) the article mainly studies the spatial pattern and influencing factors of tourism development, and it is completely reasonable to use tourism revenue as the dependent variable. (2) We seldom find that the tourism revenue is converted into the size of the district or as a share in total revenues. However, some scholars will carry out such transformation when studying the relationship between regional tourism development and economic development.

Q2: Cartogram maps are used for relative values, diagram maps are used for absolute values. No units of value.

We are very grateful for your reminder. We checked Fig2 carefully and found that it lacks units, so we redrawn Fig 2 and added units to the figure.

Q3: Lack of information on the tourist infrastructure in the districts.

In our previous literature research, we found that the expression of tourism infrastructure has many elements, such as road kilometers, the number of tourist toilets, night entertainment facilities, star-rated hotels and travel agencies. However, as an international metropolis, Beijing has relatively complete infrastructure, so there is no huge difference in public facilities such as highway mileage, toilets and traffic accessibility in various regions. The major differences are the distribution of star-rated hotels and travel agencies. At the same time, in view of the availability of data, we chose the number of star-rated hotels and travel agencies.

Thanks again. If there are suggestions for further revision, we will do our best to make the revision.

Best wishes,

Cao Yajun, Liu Jianguo

Reviewer 3 Report

The article discusses the interesting topic of spatial differentiation of tourism in Beijing. The literary research uses mainly domestic sources, but given the nature of the topic, this is still acceptable. Subsequently, the methodology is correctly designed and the results are presented.

However, I fear that the authors have an error in their results, namely in Figure 3 where they present the spatial autocorrelation results. First, the completely identical result for the years 2011, 2013, 2015, and 2019 is at least suspicious, as Figure 2 shows much greater diversity between years. Second, a completely different result in 2017 seems unbelievable, especially since the authors do not explain this exceptional situation in the commentary (so I assume that nothing extraordinary happened). And third, the categories in the image should always be shown in the same color (so don't use dark red once for the high-high category and another time for the high-low category). Overall, I think the authors should check that Local LISA is indeed entered correctly, or better explain these unusual results.

Furthermore, I found a typo in the word "Tab. 2".

Overall, the article is interesting and will certainly find its readers, but the authors need to double-check that the results presented in Figure 3 are correct.

Author Response

Dear reviewer and editor:

Thank you very much for reviewing our paper. On the basis of carefully reading the review comments, we have made the following modifications:

Q1: In Figure 3 where they present the spatial autocorrelation results. First, the completely identical result for the years 2011, 2013, 2015, and 2019 is at least suspicious, as Figure 2 shows much greater diversity between years. A completely different result in 2017 seems unbelievable, especially since the authors do not explain this exceptional situation in the commentary (so I assume that nothing extraordinary happened).

We have carefully compared the results of Fig 3. and found some small differences. We redrawn the picture and explained the results as follows:

Fig 3. shows that: during the study period, (1) the core districts and the urban function expansion districts of Beijing are basically in a high-high agglomeration state, the new urban development districts are basically in a high-low agglomeration state, and the ecological conservation districts are mostly in a low-low agglomeration state. The the core districts and urban function expansion districts of Beijing are rich in tourism resources and perfect infrastructure. They are also the gathering place of many 5A scenic spots, so they have attracted many tourists and created higher tourism income; While the new urban development districts is close to the main urban area and has relatively perfect infrastructure, but the ecological conservation districts are located in the periphery of Beijing, mostly in mountainous areas, with fewer road construction, travel agencies and star hotels, and lack of sufficient publicity. Therefore, they are not attractive to tourists, and their tourism income is relatively small. (2) The spatial pattern of tourism development in Beijing is basically stable, but in 2015 and 2017, Chaoyang District changed from the original high-high agglomeration state to high-low agglomeration state, and Shunyi district changed from low-low agglomeration state to low-high agglomeration state.The main reason for this phenomenon is that under the new tourism planning, the road construction and ecological construction in Shunyi District have promoted the rapid development of business tourism and leisure tourism, and the tourism income is higher than that of the surrounding areas.

Q2: The categories in the image should always be shown in the same color (so don't use dark red once for the high-high category and another time for the high-low category).

We thank you very much for your reminder, and carefully checked Fig 3. We found that there was indeed a problem of color inconsistency, so we redrawn the picture.

Once again, thank the reviewers and the editorial department for their valuable comments on this article. If there are suggestions for further revision, we will do our best to make the revision.

Best wishes ,

Cao Yajun, Liu Jianguo

Round 2

Reviewer 1 Report

Dear authors,

First of all, congratulations for your work. However, manuscript should be modified following considerations described as follows:

Lines 58-96. Please divide text into paragraphs and switch all the text to a “2. Literature Review” section. Introduction only must have a brief justification of the topic and the aim(s) proposed.

Lines 135-186. There is a misunderstood here. You must cite equations in text body always you refer to: (1), (2), (3) and (4). For example, in text body you must say explicitly in Equation 1 …, in Equation 2…, and so on. I am not referring to cite formula’s author. Please add explicit reference to the equations and let citations to formula’s author at it is also correct.

Lines 453-460. Please cite and reference those previous studies you are referring to.

Good luck.

Author Response

Dear reviewer and editor,

Thank you very much for your suggestions. These suggestions have important guiding significance for our article. On the basis of carefully reading the review comments, we have made the following modifications.

Q1: Lines 58-96. Please divide text into paragraphs and switch all the text to a “2. Literature Review” section. Introduction only must have a brief justification of the topic and the aim(s) proposed.

According to your suggestion, we added the second part to the article and used the second level title to make our literature review more clear. At the same time, we made clear our topic and research aims in the first part. For convenience, we use yellow in the text.

Q2: Lines 135-186. There is a misunderstood here. You must cite equations in text body always you refer to: (1), (2), (3) and (4). For example, in text body you must say explicitly in Equation 1 …, in Equation 2…, and so on. I am not referring to cite formula’s author. Please add explicit reference to the equations and let citations to formula’s author at it is also correct.

 According to your suggestion, we quoted equation (1), (2), (3) and (4) in the text, and for your convenience, we marked them in yellow.

Q3: Lines 453-460. Please cite and reference those previous studies you are referring to.

We have added the literature on the previous scholars' research results in the article, and for your convenience, we use yellow to mark it.

On the basis of completing the revision of the above problems, we have also revised and polished the language of the article to ensure that our article is more in line with the requirements of this journal.

Thanks again. If there are suggestions for further revision, we will do our best to make the revision.

Best wishes .

Cao Yajun, Liu Jianguo

Reviewer 2 Report

The article after the introduced corrections is suitable for publication

Author Response

Dear reviewer and editor,

Thank you very much for your suggestions. These suggestions have important guiding significance for our article. On the basis of carefully reading the review comments, we have made the following modifications.

Q1: The article after the introduced corrections is suitable for publication

According to your suggestion, we have revised and improved the language and text of the article, and carefully checked it to ensure that it is more in line with the requirements of this journal.

Thanks again. If there are suggestions for further revision, we will do our best to make the revision.

Best wishes.

Cao Yajun, Liu Jianguo
